# Influence of Ceria Addition on Crystallization Behavior and Properties of Mesoporous Bioactive Glasses in the SiO_2_–CaO–P_2_O_5_–CeO_2_ System

**DOI:** 10.3390/gels8060344

**Published:** 2022-05-31

**Authors:** Elena Maria Anghel, Simona Petrescu, Oana Catalina Mocioiu, Jeanina Pandele Cusu, Irina Atkinson

**Affiliations:** Institute of Physical Chemistry ‘IlieMurgulescu’ of Romanian Academy, Splaiul Independentei 202, 060021 Bucharest, Romania; manghel@icf.ro (E.M.A.); omocioiu@icf.ro (O.C.M.); jeaninamirea@yahoo.com (J.P.C.)

**Keywords:** sol–gel processes, spectroscopy, X-ray methods, thermal properties, bioactive glass, silicate, biomedical applications

## Abstract

Knowledge of the crystallization stability of bioactive glasses (BGs) is a key factor in developing porous scaffolds for hard tissue engineering. Thus, the crystallization behavior of three mesoporous bioactive glasses (MBGs) in the 70SiO_2_-(26-x)CaO-4P_2_O_5_-xCeO_2_ system (x stands for 0, 1 and 5 mol. %, namely MBG(0/1/5)Ce), prepared using the sol–gel method coupled with the evaporation-induced self-assembly method (EISA), was studied. A thermal analysis of the multiple-component crystallization exotherms from the DSC scans was performed using the Kissinger method. The main crystalline phases of Ca_5_(PO_4_)2.823(CO_3_)_0.22_O, CaSiO_3_ and CeO_2_ were confirmed to be generated by the devitrification of the MBG with 5% CeO_2_, MBG5Ce. Increasing the ceria content triggered a reduction in the first crystallization temperature while ceria segregation took place. The amount of segregated ceria of the annealed MBG5Ce decreased as the annealing temperature increased. The optimum processing temperature range to avoid the crystallization of the MBG(0/1/5)Ce powders was established.

## 1. Introduction

Since their discovery in the late 1960s, bioactive glasses have been intensively studied due to their excellent bioactive response in hard tissue engineering [1]. However, the main limitation of the use of bioactive glass (BG) in obtaining porous scaffolds that mimic the structure of human bones [2] consists of improper mechanical characteristics, especially brittleness. Additionally, crystallized glass-ceramics show a lower surface reactivity in physiological solutions as a consequence of the reduction in the surface Si-OH linkages in comparison with the glassy counterparts [3]. Both brittleness and bioactivity are influenced by the crystallization (devitrification) behavior of BGs [1]. To overcome the limitation of BG crystallization during scaffold preparation [2], few solutions are used, e.g., tailoring BG composition, sol–gel preparation, understanding crystallization behavior and obtaining polymer-bioactive glass composites. The silica content, type of glass modifier (Na, K, Ca, Mg and Ba [4]) and doping oxides (ZnO, Ce_2_O_3_, Ga_2_O_3_, Bi_2_O_3_, Nb_2_O_5_ [5,6,7,8], etc.) are critical factors determining the ability of BGs to crystallize [9,10]. Although BGs with a silica content of up to 80% are still bioactive [11], they are denser than the Hench’s 45S5 Bioglass^®^ with 45% SiO_2_ [1]. A better workability was reported for alkali-free BGs developing wollastonite (CaSiO_3_) during crystallization in comparison with sodium–calcium–silicate phases, such as combeite (Na_2_Ca_2_Si_3_O_9_) [12,13]. To inhibit the crystallization of sodium–calcium–silicate phases in 45S5 while preserving their sintering and fiber-drawing abilities, magnesium and zinc were partially substituted for calcium [14]. The latter ions enabled the processing range to be widened, namely, the temperature range within the glass transition (T_g_) and crystallization onset, T_x_. Except for BG formulation, the synthesis method highly influences the surface area and pore architecture, which are essential for an adequate surface reactivity in physiological fluids required by scaffolds in bone regeneration [2,15,16]. In contrast with melt and sol–gel-derived BGs, an enhanced bioactivity of the MBGs obtained by the sol–gel method coupled with the surfactant method is influenced by their pore architecture [15,16,17,18,19]. The ability to form ordered mesopores (2–50 nm sized pores) is mainly affected by the SiO_2_ content and selected surfactant [15]. The most well known method for tuning the pore architecture of a BG obtained using sol-gel processing is the evaporation-induced self-assembly method (EISA) [16,17]. No correlation between the ordered mesoporosity and devitrification tendency has been previously reported in the literature.

Differential thermal analysis (DTA) and differential scanning calorimetry (DSC) are useful techniques for studying the crystallization of BGs [1,9], while the kinetic analysis of the thermal data provides information on the reactivity and stability of BGs. Although richer silica BGs do not easily crystallize [11,20], the few kinetics reports on crystallization in BGs are mostly conducted on the 45S5 Bioglass^®^ [1,12,21]. In comparison to richer-silica BGs 1-98(53SiO_2_-22CaO-6Na_2_O-11K_2_O-5MgO-2P_2_O_5_-1B_2_O_3_, wt.%) and 13-93(53SiO_2_-20CaO-6Na_2_O-12K_2_O-5MgO-4P_2_O_5_, wt. %) with surface nucleation [11], the crystallization of the 45S5 Bioglass^®^ suddenly proceeds from the surface to bulk phase [1]. The kinetics of the thermally simulated devitrification of BGs and their corresponding energy barrier has been very often studied by using the Kissinger method for the n^th^ order reactions [22,23].

Since the thermal behavior of MBGs obtained using sol-gel process helps when choosing parameters for fiber and bioactive scaffold preparation, this work presents the non-isothermal crystallization kinetics of Ce-containing MBGs in the 70SiO_2_-(26-x)CaO-4P_2_O_5_-xCeO_2_ system (x stands for 0, 1 and 5 mole %) by using DSC data. The identification of the crystallization mechanism and crystalline phases was assessed.

## 2. Results and Discussion

### 2.1. Phase Evaluation in the G(0/1/5)Ce Gels

The FT-IR spectra of the dried G(0/1/5)Ce gels, which are the un-stabilized MBGs, as illustrated in Figure 1a and exhibited characteristics of a silicate structure at 473, 825 and 1049 cm^−1^ due to the rocking, bending and stretching modes of the Si-O-Si bonds that formed into coalesced silica particles during condensation processes. The band located at ~570 cm^−1^ [24] might be assigned to four-fold rings, i.e., Si(OSi)_3_(OR) (R stands for C_2_H_5_ or H) and/or four-fold silanol rings, as well as to the stretching and bending modes of the P-O bonds. Nitrate ions, depictable by the intense band at 1380 cm^−1^ and smaller bands at 822 and 740 cm^−1^ [25], indicated the non-incorporation of calcium ions into the silica network [26]. Calcium nitrate, covering silica nanoparticles, was reported to exist in dried gels of 70S30C (70SiO_2_-30CaO, mol. %) up to 350 °C [26,27]. Hence, the IR spectra of the dried 70S30C gels resembled those of calcium nitrate, but the 1047 cm^−1^ band was thinner for the latter compound, and the shoulder at approximately 1080 cm^−1^ indicates the formation of Si-O-Si [27]. The coexistence of the 1049 and 1079 cm^−1^ spectral features is easier to observe for the G5Ce sample in Figure 1a. The O-H presence in H_2_O and alcohols [24,28] was indicated by the 1635 and 3423 cm^−1^ bands. The small band at approximately 950 cm^−1^ of the G5Ce spectrum was due to Si-OH linkages [28]. The organic residue was identified by the C-H stretching modes of the CH_2_ and CH_3_ groups at 2938 and 2878 cm^−1^ [28].

UVRaman spectroscopy, enabling fluorescence avoidance, as well as the selectively enhanced detection of nitrates [29], was used for the first time, in the present study, to investigate the surface of the cerium-doped gels in the CaO-SiO_2_-P_2_O_5_ system. The ν_1_(NO_3_^−^) band [30] in Figure 1b (inset) was up-shifted for the G(1/5)Ce in comparison with that for cerium-free gel, which is very likely due to hydrated water and metal cation (calcium and cerium cations) effects [31]. This behavior confirmed a lack of calcium nitrate incorporation, as already depicted by IR, as well as the lack of cerium nitrate, which covers the already formed Si(OSi)_3_(OR) network. Additionally, hydroxyl from molecular H_2_O (1600–1650 cm^−1^ and 3350–3500 cm^−1^ ranges), organic residue (symmetric and asymmetric C-H stretching modes of the methylene, CH_2_, and methyl, CH_3_, within 2600–3100 cm^−1^) and the silicophosphate network of four-fold rings at approximately 490 cm^−1^ [24,32] are depicted in Figure 1b. Bands at approximately 1415 and 1460 cm^−1^ are attributable to bending vibrations of the CH_2_ and CH_3_ groups [32]. The tinny band at 1140 cm^−1^ indicated the formation of the Si-O-P bonds [32]. Symmetric vibrations of the P=O [32] were observed at 1279 cm^−1^. In order to stabilize these dried gels, namely, to remove the organic and nitrate residue, their thermal behavior should be assessed.

### 2.2. DTA/DTG/TG Analysis of the G(0/1/5)Ce Gels

The DTA/DTG/TG curves of the G(0/1/5)Ce materials are illustrated in Figure 2a,b. The TG curves in the inset of Figure 2 indicate the lowest thermal stability of the G1Ce sample up to 160 °C, intermediate behavior over the 160–250 °C range, while above 250 °C, its stability improved more than the other two gels. Hence, the rate of the mass loss was not solely dependent on the gel composition for the whole temperature range investigated. The overall mass loss ranged from 58.39% (G1Ce) to 60.80% (G5Ce). The first stage, at approximately 70 °C, with a mass loss <8wt. %, for all the gels investigated, corresponded to the physically adsorbed water [33,34,35] and ethanol [32,36]. The next two stages (Table 1) record a mass loss of a maximum of 37%, up to 300 °C, corresponded to the vaporization o water and decomposition of organic residue. Chemisorbed water resulted from precursor condensation was removed at approximately 230 °C [13]. The third stage was due to the alkoxy group decomposition [37,38].

Above 100 °C, the weight loss stages were accompanied by two exothermic events on the corresponding DTA curves (Figure 2b). The second DTA exotherm had left and/or right sided shoulders due to the complex decomposition process. Ethyl groups of the TEOS and TEP along with un-hydrolyzed precursor vaporization [32] were responsible for the first exothermic event at approximately 200 °C. 

The deegradation of Pluronic P-123 and nitrate by-products [26] accounted for the last (fourth and fifth) mass loss stages. The removal of the nitrate byproducts confirmed incorporation by the diffusion of the calcium [26] and cerium ions into the silica network. The calcium diffusion process in the 70S30C gels takes place above 400 °C. Thus, the thermal events at 531 and 608 °C, of the DTG curve for the G1Ce in Figure 2a, correspond, are very likely due to loss of the remnant surfactant and nitrates, respectively [26,35]. A tinny endotherm event at 600 °C was present on the DTG curve of the G5Ce (Figure 2a).

According to the thermal features above, the obtained gels were two-step thermally treated to obtain MBGs. The first step at 300 °C (1 h) was required for water and organic residue removal, while the second one was carried out at 700 °C (3 h) when no mass loss occurred and calcium and cerium incorporation in the phosphosilicate network took place. To evaluate the stabilization efficiency of the Ce-containing MBGs under discussion, FT-IR and XRD measurements were carried out. The XRD patterns of the MBG(0/1/5)Ce samples calcined at 700 °C [33] showed the presence of halos specific to the glassy phase. 

### 2.3. Phase Identification in the Devitrified Ce-Containing MBGs

The DSC curves of the stabilized BGs powders, MBG(0/1/5)Ce, collected by heating with a 10 °C/min. rate (Figure 3a), had multiple crystallization exotherms. Thus, two crystallization effects were recorded for MBG1Ce, while three components were required for MBG(0/5)Ce (Figure 3b). The peculiar behavior of MBG1Ce regarding higher pore interconnectivity was determined [33] from the wider hysteresis loop of the nitrogen adsorption/desorption isotherms of the MBG(0/1/5) samples. Hence, except for MBG formulation, the textural data can influence thermal behavior. Three exotherms of crystallization were also reported for the 70S26C4P (70SiO_2_-26CaO-4P_2_O_5_, mol. %) BG obtained using the sol–gel method in the absence of surfactants and stabilized in a single step at 700 °C [36]. Decreasing temperature for the first crystallization exotherm in Figure 3a was noticeable as ceria content increased. Jones et al. [39] reported a crystallization peak at 873 °C for β-wollastonite in the foamed and unfoamed monoliths of 70S30C, while Sigueira and Zanotto [36] found apatite in primary phase crystallizing at 900 °C. In the case of the MBG0Ce powders, the first exotherm of crystallization (onset temperature, T_x_, of 863 °C and peak temperature, T_c1_,of 878 °C) was assignable to an apatite phase (XRD data in Figure 4a and Table 2), whereas the other two exotherms belonged to wollastonite (β-phase) and higher temperature wollastonite, pseudowollastonite or α-wollastonite [25]. The crystallization of wollastonite phases was accompanied by the enhancement of mechanical properties [40].

The formation of the Ca_5_(PO_4_)_2.823_(CO_3_)_.22_O (64.45% crystallinity) was noticeable in the X-ray pattern of the MBG1Ce isothermally treated at 865 °C (MBG1Ce_865 in Table 2), which is similar to the MBG0Ce_878 sample (Figure 4b). Hence, the annealed samples at the first crystallization effect, MBG(0/1)_T_c1_, showed the presence of an apatite phase, although only amorphous phases were shown prior to thermal treatments (Figure 4b). Ceria prevailed in the annealed MBG5Ce_T_c1_(Figure 4b and Table 2). The annealing of the MBG(0/1/5)Ce BGs at approximately 870 °C (Table 2) induced the formation of apatite phase.

Crystalline phosphates were observed in the IR spectra of the two annealed BGs with three crystallization effects, MBGS(0/5)_(T_c1_/T_c2_/T_c3_), as shown in Figure 5, due to its bands at ~570 and 609 cm^−1^ as a result of the P-O bending vibrations of the orthophosphate PO_4_^3−^ groups [39,40]. These bands were used to monitor glass bioactivity [41] since the other P-O vibrations overlapped with the Si-O vibrations. Only the 565 cm^−1^ band was observed for the MBGS1Ce_(Tc1/TC2) in Appendix A. A small band at ~428 cm^−1^ might have belonged to Ce-O [42] in MBG5Ce_(830/876). The asymmetric Si-O-Si stretching mode at ~1092 cm^−1^ [43], and symmetric stretching and bending bands at 794 cm^−1^ and 465 cm^−1^ were also present in the IR spectra of all the thermally treated MBG0Ce as shown Figure 5a. Moreover, IR bands located at 1018, 937, 903, 720, 684, and 642 cm^−1^ of the samples treated at a higher temperature, MBG0Ce_(951/1012), belonged to β-wollastonite [44]. IR data reported for the sol–gel 70S26C4P bioglass annealed at 1000 °C [44] revealed the formation of quartz (798/780 doublet and 697 cm^−1^). Quartz absence in the XRD pattern of MBG0Ce_1012 might be due to its low quantity under the XRD detection limit. Barely perceptible shoulder at 984 cm^−1^ for the MBG0Ce_1012 sample belonged to pseudowollastonite (PW). This finding, supported by the XRD pattern in Figure 5a, was also reported for the sintered BG of 58S (58 wt. % SiO_2_, 33 wt. % CaO and 9 wt. % P_2_O_5_) at 1100 °C [44].

Raman spectroscopy is a powerful technique for studying nucleation and growth in annealed glasses [45]. Moreover, UV-Raman spectroscopy accesses the surface information of BGs. The annealed MBGS5Ce_T samples showed distinct Raman features below 750 cm^−1^ (Figure 6 and Appendix A). Thus, the band at 438 cm^−1^ of the MBG5Ce_830 is attributable to rocking modes of the bridging oxygen atoms located perpendicular to the P–O–P plane and of the F_2g_ modes of CeO_2_ [46]. More intense bands of defects, D_1_ and D_2_, of the MBG5Ce_(830/876) samples in comparison with commercial ceria were due to oxygen vacancies associated with Ce^3+^ and Ce^4+^ sites [46]. The defect bands were either thermally and/or dopant activated [46]. The overtones (2LO) of ceria at approximately 1167 cm^−1^ were also shown for the first exotherm-annealed MBG5Ce_(830/876). The symmetric stretch of the Q^0^(P) units (595 cm^−1^) [8] indicated the formation of a crystalline phosphate phase in the MBG5Ce_(RT/830/876). The band at 464 cm^−1^ [47] revealed the presence of quartz in the MBG5Ce_906 sample. The wollastonite band (Si-O bending [48]) at approximately 635 cm^−1^ (Si-O-Si bonds [8] in Q^2^(Si)) supported by XRD and IR data for the second exotherm-annealed sample MBG5Ce_876. A band at 620 cm^−1^ was reported [49] for cerium phosphate. UV-Raman spectra of the annealed MBGS5Ce_T sample showed the ν_1_(PO_4_^3−^) band at approximately 950 cm^−1^, as observed in Figure 6 [24,45]. The same band could also belong to the Q^2^(Si) units (SiO_4_ tetrahedra with two non-bridging oxygen atoms, NBOs) of the metasilicate chains (Si_2_O_6_^2−^) [24]. The abundance of the Q^2^(Si) units influenced the glass bioactivity. The band at approximately 846 cm^−1^, which was seen in all the samples (Figure 6), corresponds to the Q^0^(Si) units that provided information about the devitrified glass structure.

### 2.4. Crystallization Behavior of the Stabilized Ce-Containing MBGs

To obtain workability information (processing range, T_x_-Tg, reactivity, and stability) of the MBGs kinetic approach of the DSC runs were collected at various rates (Figure 7). Although glasses can show multiple crystallization exotherms only the first one was used in the calculation of glass stability defined as the crystallization resistance of glass upon heating [9]. The most simplified estimation of the glass stability is to obtain the T_x_-T_g_ quantity from DSC measurements [1], i.e., temperature range where crystallization is avoided. A value of higher than 100 °C of this quantity implies good thermal stability, which was the case for all MBGs presented here (Figure 7 and Appendix A). The lowest T_x_-T_g_ value was recorded for the MBG5Ce (149 °C at 10 K/min.). Other parameters (Hruby, Weinberg, etc.) for measuring glass stability imply the use of glass melting temperature [9]. However, the measurement of the melting temperature of the broad endothermal DSC effect is rather inaccurate.

Therefore, considering crystallization exotherm dependence on the DSC heating rate, activation energy (E_a_) was derived using the Kissinger equation (Equation (1)) for the first exothermof crystallization. Plots of [−ln (β/T_C1_)] as functions of [1000/T_C1_] for the first crystallization exotherm of the MBG(0/1/5)Ce samples are presented in Figure 8. The straight lines used to fit these plots represent E_a_/R activation energy for crystallization and gas constant. The high activation energy of apatite crystallization in MBG(0/1)Ce samples (Table 3) indicates the good thermal stability of the two samples. Larger E_a1_values of the MBG(0/1)Ce than those reported for coarse and fine apatite particles, 514 and 482 KJ/mol K, crystallized in lime- and magnesia- containing silicate glasses [40], indicate a lower tendency for the crystallization of these MBGs. The Avrami exponent, n, calculated using Equation (2) indicates the surface crystallization of the ceria (MBG5Ce) and apatite phase (MBG(0/1)Ce). As well as size, the shapes of the glass particles influence crystallization kinetics [9]. Thus, the wide FWHM of the first crystallization exotherm (~40 °C in Appendix A) for the MBG5Ce might originate from spherical and/or cuboid glass particles, whereas much narrower MBG(0/1)Ce exotherms (~20 °C) can be due to prolate and needle-like apatite particles [9]. Smaller E_a_ and T_x_-T_g_ values were obtained for the MBG5Ce compared with MBG(0/1)Ce samples. Surface area values, S_BET_, as reported elsewhere [33], decreased in the same succession as E_a_ for the first crystallization, namely, MBG0Ce (307 m^2^/g) > MBG1Ce(230 m^2^/g) > MBG5Ce (223 m^2^/g), and inversely proportional to the increase in ceria content.

Given the distinct thermal behavior and segregation of CeO_2_ in the MBGS5Ce the bioactivity study focused on this composition.

### 2.5. Bioactivity of the MBG5Ce_T

Multiple steps were identified in developing a hydroxy-carbonate apatite (HCA) layer on the surface of BGs when treated with simulated biological fluid, SBF [1]. This complex process of obtaining an HCA layer that is well-matched with natural bones and teeth relies on three stages: leaching, dissolution, and precipitation. The leaching of Ca^2+^ from BGs and exchanging with H^+^ and H_3_O^+^ from SBF was followed by the dissolution process when the breaking the Si-O-Si bonds caused the formation of Si-OH at the surface and as well as the release of Si(OH)_4_ into SBF. Superficial Si-OH underwent polycondensation into a silica gel layer. Glass-released and SBF-originating calcium and phosphate ions migrated to the silica gel layer and precipitated as an amorphous Ca-P-rich layer which subsequently crystallized into HCA due to carbonate incorporation. The dissolution process is highly dependent on glass and/or glass-ceramic composition, structure, and morphology. For instance, Ce^3+^ ions released by the glass surface can compete with Ca^2+^ for phosphate ions in SBF to form insoluble CePO_4_ and hence delay HCA formation [50,51].

The surface reactions of the MBG particles can be also monitored using UV-Raman spectroscopy with a low penetration depth. To assess mineralization ability, SEM, UV-Raman, and IR investigations were carried out after 14-day immersion of the annealed MBG5Ce_(RT/830/876) samples in SBF. The FTIR spectrum shown in Figure 9a of the 14-day soaked MBG5Ce_830 in SBF was dominated by the 1060 and 466 cm^−1^ bands of the Si-O-Si stretching, and bending modes. The lack of bands at approximately 602 cm^−1^ for the MBG5Ce_(830/875)_14 samples showed either that the crystalline phosphate phase existing in these samples prior to immersion was soluble in SBF or an amorphous phosphate phase was formed, despite the segregation of CeO_2_ (see XRD data in Figure 4b and Table 2) in the MBG5Ce sample, which is known to hinder bioactivity [33,52]. IR and Raman spectra of the sample without thermal treatment indicated the formation of HCA (intense peak of PO_4_^3−^ and CO_3_^2−^ at 954 and 1100 cm^−1^ [41,51,52] in Figure 9b and 602 cm^−1^ in Figure 9a) after 14 days of immersion in SBF (MBG5Ce_RT_14). Instead, the wide band peaking at approximately 1200 cm^−1^ (Figure 9b) in the MBG5Ce_(830/876)_14 samples provided evidence of ceria. Except fluorescence removal, UV-Raman spectroscopy is a very sensitive technique to identify the defective structure of ceria at a lower penetration depth.

The silica gel layer obtained in the early stages of the bioglass immersed in SBF [38] was subsequently penetrated by calcium and phosphate ions and an amorphous calcium phosphate layer covered the sample surface (Stage 4) [53]. Hydroxyl, phosphate, carbonate and calcium ions formed an outer hydroxyapatite layer.

Delayed bioactivity was confirmed by IR, UV-Raman and SEM data (Figure 10) collected on the annealed MBG5Ce at the second crystallization event corresponding to wollastonite crystallization. Shoulders at approximately 642 and 953 cm^−1^ of the 14-day soaked MBG5Ce_875 spectrum indicated β-wollastonite phase.

The SEM micrograph of the sample without annealing treatment immersed for 14 days in SBF, MBG5Ce_14, showed the formation of hydroxyapatite. Moreover, its Ca/P ratio of 1.78 (EDS spectrum illustrated in the inset of Figure 10b) was closer to that of the natural hydroxyapatite (1.67). Cerium and phosphorous depicted in the inset of Figure 10 confirmed the presence of segregated ceria and apatite phase crystallized at the first crystallization peak of the MBG5Ce_830 prior to immersion in SBF. Tinny needle crystals on the surface of theMBG5Ce_830_14 sample might point out incipient apatite phase formation. Conversely, the MBG5Ce_876_14 micrograph (Figure 10d) is evidence of the prevention of hydroxyapatite formation in samples containing crystallized wollastonite at the surface. The highest Ca/P ratio of 3.54 is further proof of this.

Delayed bioactivity was confirmed by IR, UV-Raman and SEM data (Figure 10) collected for the annealed MBG5Ce at the second crystallization event corresponding to wollastonite crystallization. Shoulders at approximately 642 and 953 cm^−1^ of the 14-day immersed MBG5Ce_875 spectrum indicated β-wollastonite phase.

## 3. Conclusions

Information on the crystallization avoidance, i.e., thermal stability and temperature processing window (Tx-Tg range) of some cerium-containing MBGs, MBG(0/1/5), was obtained in this study. Double and triple crystallization events of the DSC curves were recorded for the MBG(0/1/5)Ce powders. The XRD findings of the annealed samples at the crystallization peak temperatures revealed the crystallization of apatite, wollastonite, and ceria phases. The large activation energy (>500 KJ/mol) corresponding to the first crystallization event, derived by using the Kissinger method, indicated a low crystallization tendency which is appropriate for the thermal processing of the MBGs. The addition of 5% ceria to the MBGs, MBG5Ce, caused the lowering of the first crystallization exotherm from 876 °C to 830 °C and the narrowing of the processing window to ~100 °C while ceria segregation took place. Conversely slower bioactivity was seen for the annealed MBG5Ce at its first crystallization exotherm. According to these results, the MBG(0/1/5)Ce powders presented here are suitable for the production of porous biomedical scaffolds. Further cellular tests are nedeed to investigate the biological response of the annealed MBGs.

## 4. Materials and Methods

### 4.1. Materials

Tetraethylorthosilicate (TEOS of >98% from Merck, Darmstadt, Germany), pluronic^®^ P123 (Sigma-Aldrich, Darmstadt, Germany)), triethylphosphate (TEP of 99% from Aldrich), calcium nitrate tetrahydrate (99% p.a., Carl Roth, Karlsruhe, Germany) and cerium nitrate hexahydrate (99%, Aldrich) were the starting materials. Cerium oxide (99.9% from Loba-Chemie, Mumbai, India) was used for comparison.

### 4.2. Sol–Gel Preparation Methods

The sol–gel synthesis coupled with the evaporation-induced self-assembly method (EISA) using Pluronic^®^ P123 as a structure-directing agent was employed to obtain cerium-doped MBGs in the 70SiO_2_-(26-x)CaO-4P_2_O_5_-xCeO_2_ system (x stands for 0, 1 and 5 mole %) as described elsewhere [33]. The obtained dried gels, denominated G(0/1/5)Ce gels according to the molar percent of CeO_2_, were two-step thermally treated according to the thermal effects indicated by the corresponding TG/DTG/DTA curves in Figure 2.

Further, the glass powders were isothermally crystallized separately in a Pt crucible in air at the temperature (T) corresponding the DSC exotherms of crystallization, MBG(0/1/5)_T for 3h and 24 h, respectively.

### 4.3. Bioactivity of the Devitrified Glasses

The in vitro bioactivity of the devitrified MBG(0/1/5)_T powders was checked in simulated body fluid (SBF) for 14 days, as described elsewhere [5].

### 4.4. Characterization

#### 4.4.1. Thermal Characterization

The thermal behavior of the G(0/1/5)Ce gels was determined through differential thermal analysis and thermo-gravimetric analysis using Mettler Toledo TGA/SDTA 851e equipment, in Al_2_O_3_ crucibles and in a flowing air atmosphere. The maximum temperature was set at 1000 °C and the heating rate was 10 °C/min.

Thermal stability analysis of the calcined MBGs, MBG(0/1/5)Ce, was performed by means of a LabSysEvo in a platinum crucible in an argon atmosphere. Approximately 26 mg of grounded glass with a particle size below 75 μm was used for DSC measurement within 25–1200 °C. Crystallization kinetics analysis was conducted with heating rates, β, of 5, 10, 15, 20 and 25 °C/min. after appropriate equipment calibration for each rate with In, Sn, Zn, Al and Ag. The glass transition temperature (T_g_) established based on the inflection point and peak temperature of crystallization (T_p_) were determined by using Calisto 1.051 software from the DSC curves of the respective glasses.

The activation energy (E_a_) was calculated using the Kissinger equation [22,23] according to the DSC data:(1)$$\ln\left(\frac{\beta }{T_{C}^{2}}\right)=\mathrm{constant}-\frac{E_{a}}{{\mathrm{RT}}_{C}}$$
where: T_c_ is crystallization temperature measured at various heating rates, β, and R is the gas constant. The activation energy, E_a_, for crystallization is derived from slope (−E_a_/R) of the straight line of the ln(β/T_C_^2^) versus (1000/T_C_) representation.

The Avrami exponent (n), an indicator of crystal growth dimensionality, was determined using Equation (2) proposed by Augis and Bennett [54,55]:(2)$$n=\frac{2.5}{\Delta T_{\mathrm{FWHM}}}\frac{T_{C}^{2}}{\frac{E_{a}}{R}}$$
where: ΔT_FWHM_ is the full width at the half maximum of the crystallization exotherm from the DSC curve.

#### 4.4.2. Structural Characterization

The structures of the G(0/1/5) gels were investigated using IR and UV-Raman spectroscopies. Additionally, crystalline phases obtained through isothermal devitrification in accordance with the DSC exotherms of crystallization were identified by means of X-ray Diffraction (XRD), IR and UV-Raman spectroscopies. Thus, the XRD patterns of glassy and devitrified counterparts were recorded using a RigakuUltima IV diffractometer (Rigaku Corporation, Tokyo, Japan) equipped with CuKα radiation, with 2°/min and a step size of 0.02°. Fourier transform infrared (FTIR) spectra of the were recorded without additional slice preparations, in the 400–4000 cm^−1^ domain with a sensitivity of 4 cm^−1^ by using a Thermo Nicolet 6700 spectrometer (Thermo Fisher Scientific Inc., Waltham, MA, USA). UV-Raman spectra of the gels, stabilized and thermally devitrified MBGs were collected by means of a Labram HR 800 spectrometer (HORIBA FRANCE SAS, Palaiseau, France) equipped with a UV laser line (325 nm from Kimmon Koha Co., Ltd., Tokyo, Japan), grating of 2400 lines and a 40x/0.47 NUV objective, assuring a laser spot on the sample smaller than 1 μm.

The morphology and EDS characterization of the MBGs samples were carried out by using a FEI Quanta3DFEG (FEI, Brno, Czech Republic) microscope equipped with an Octane Elect EDS system. Secondary electron images were recorded at an accelerating voltage of 5 and 10 kV in high-vacuum mode. Samples were recorded without sputter coating with conductive material.

## Figures and Tables

**Figure 1 gels-08-00344-f001:**
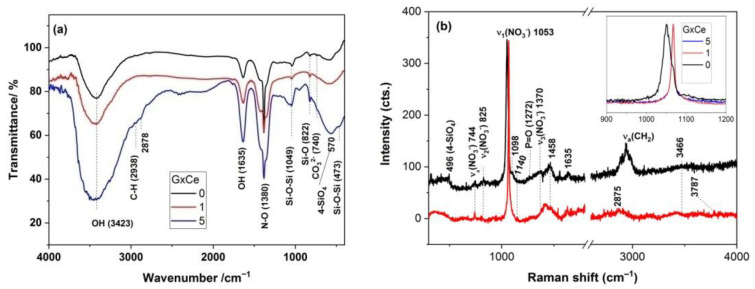
(**a**) IR and (**b**) UV-Raman spectra of the GxCe gels (x stands for 0, 1 and 5% ceria).

**Figure 2 gels-08-00344-f002:**
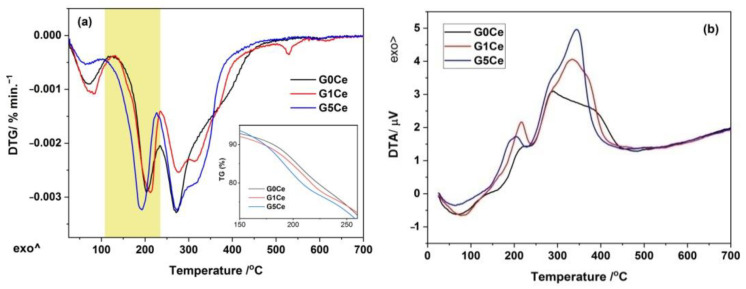
(**a**) DTG/TG curves of the G(0/1/5)Ce gels and (**b**) DTA curves of the G(0/1/5)Ce gels.

**Figure 3 gels-08-00344-f003:**
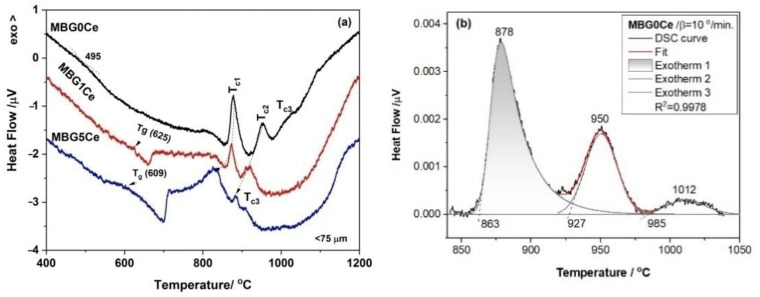
(**a**) DSC runs at a heating rate of 10 °C/min. for the stabilized MBG(0/1/5)Ce and (**b**) three-component fitted DSC curve of the MBG0Ce crystallization.

**Figure 4 gels-08-00344-f004:**
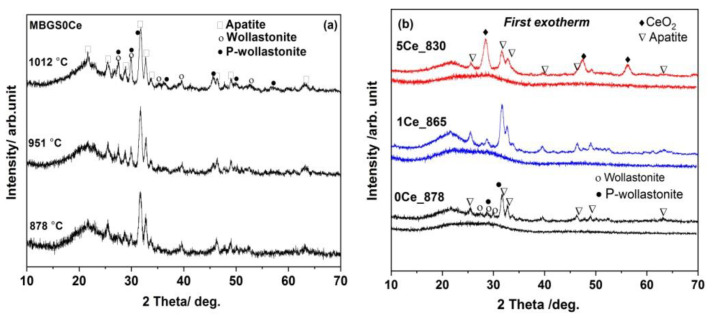
XRD patterns of the annealed: (**a**) MBG0Ce_(878/951/1012) and (**b**) MBG(0/1/5)_T_c1_ samples at first crystallization exotherm.

**Figure 5 gels-08-00344-f005:**
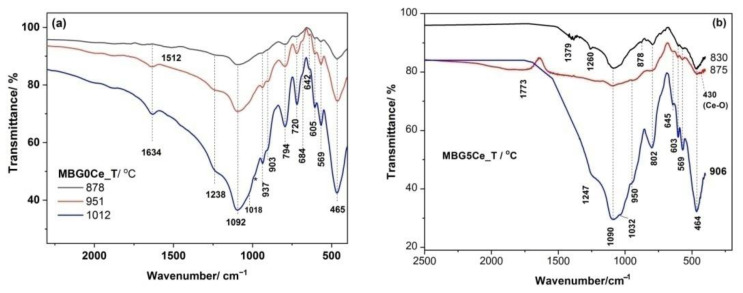
FTIR spectra of the annealed (**a**) MBG0Ce_T and (**b**) MBG5Ce_T samples at all crystallization exotherms (* stands for the 984 cm^−1^ shoulder).

**Figure 6 gels-08-00344-f006:**
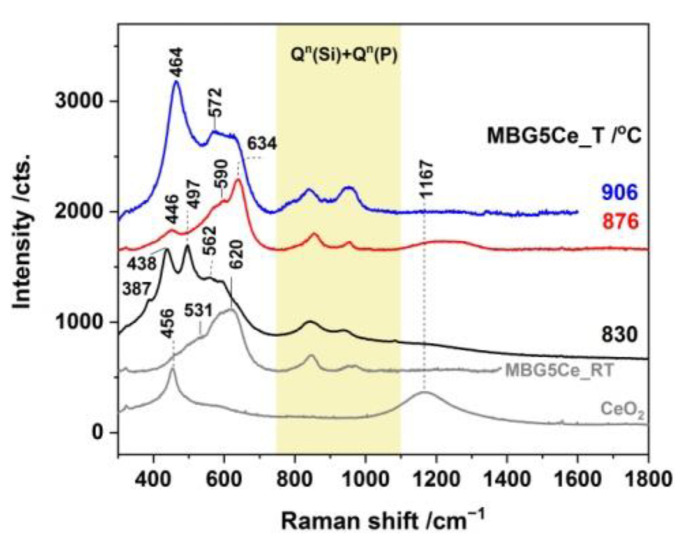
UV-Raman spectra of annealed MBG5Ce_T at crystallization exotherms and commercial CeO_2_.

**Figure 7 gels-08-00344-f007:**
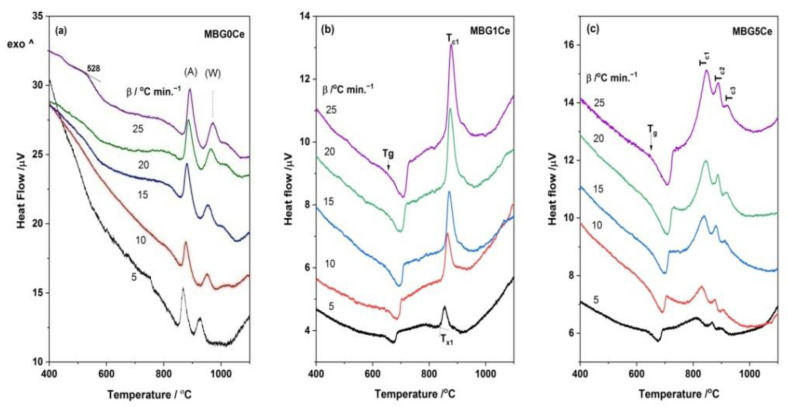
DSC runs with various heating rates of: (**a**) MBG0Ce, (**b**) MBG1Ce and (**c**) MBG5Ce (A and W stand for apatite and wollastonite).

**Figure 8 gels-08-00344-f008:**
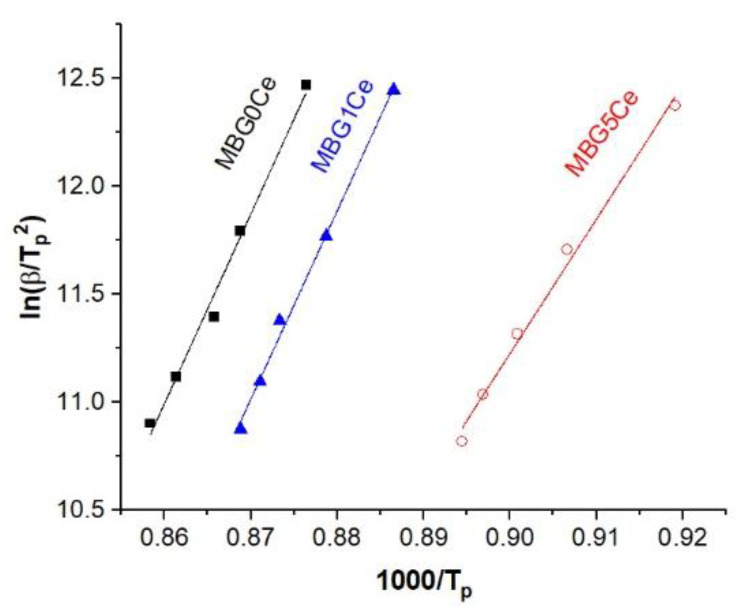
Linear fit of the Kissinger plots for the first exotherm of crystallization (T_c1_) of the MBG(0/1/5)Ce.

**Figure 9 gels-08-00344-f009:**
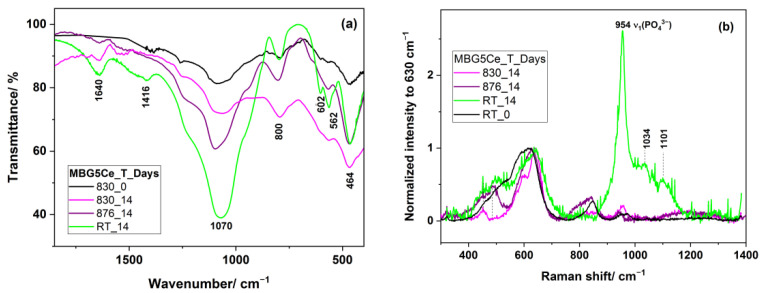
(**a**) FT-IR and (**b**) Raman of the MBG5Ce_(RT/830/875) soaked in SBF for 14.

**Figure 10 gels-08-00344-f010:**
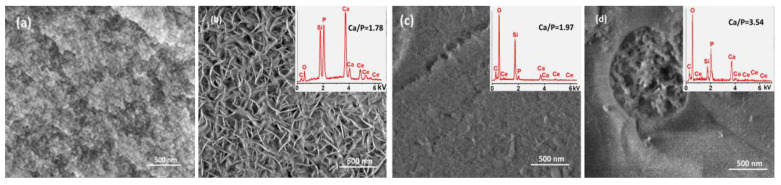
SEM images of the MBG5Ce sample: (**a**,**b**) untreated, and 14-day SBF soaked; (**c**) MBG5Ce _830_14 and (**d**) MBG5Ce _876_14.

**Table 1 gels-08-00344-t001:** Thermogravimetric decomposition data of the G(0/1/5)Ce gels.

Gel Code	Stage No	ΔT/°C	Mass/%	Assignment	Total Loss/%
0Ce	1	25–132	6.30	Physically absorbed water [34,35]	59.91
	2	132–234	16.16	Chemisorbed water and organic oxidation [13]	
	3	234–301	18.05		
	4	301–497	18.93	Pluronic and nitrate decomposition [26,35]	
1Ce	1	25–134	7.14	Physically absorbed water	58.39
	2	134–236	16.63	Chemisorbed water and organic oxidation	
	3	236–298	12.94		
	4	298–390	13.85	Pluronic and nitrate decomposition	
	5	390–640	6.96		
5Ce	1	25–106	3.39	Physically absorbed water	60.85
	2	106–227	20.17	Chemisorbed water and organic oxidation	
	3	227–293	16.11		
	4	293–620	20.04	Pluronic and nitrate decomposition	

**Table 2 gels-08-00344-t002:** List of thermal treatment holds for 24 h at DSC exotherms of crystallization and corresponding XRD data of MBGs (particle size 75 μm).

MBG Code	T_c1_ (°C/Phases)	T_c2_ (°C/Phases)	T_c3_ (°C/Phases)
0Ce	878/	951/	1012/
	A (68.01%), W (20.72%) PW (11.27%)	A (68.34%), W (10.04%), PW (21.62%)	A (70.98), W (17.69%), PW (11.33%)
1Ce	865/	900/	-
	A (90.10%), W (9.90%)	A (85.88%), W (15.12%)	
5Ce	830/	876/	906/
	A (38.21%), C (14.10%), PW (47.68)	A (54.15%, C (13.41%)	A (59.05%), C (7.80%), PW (33.14%)
		PW (32.44%)	

A = apatite (JCPDS 01-073-1731), W = wollastonite (JCPDS 00-900-8151), PW = pseudowollastonite (JCPDS 01-074-0874), C = ceria (JCPDS 00-043.

**Table 3 gels-08-00344-t003:** Linear fit of the Kissinger plots for the first crystallization exotherms of stabilized MBG(0/1/5)Ce.

Sample	Intercept	Slope	E_a_	n	R^2^
MBG0Ce	64.5969	87.8985 ± 5.22	730.82 ± 43.40	1.5979	0.9860
MBG1Ce	64.9096	87.2713 ± 3.23	725.61 ± 26.85	1.8879	0.9959
MBG5Ce	−44.7813	62.2325 ± 3.17	517.43 ± 26.35	1.1871	0.9922

## Data Availability

The data presented in this study are contained within the article.

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
