# Peer review of "Influence of Ceria Addition on Crystallization Behavior and Properties of Mesoporous Bioactive Glasses in the SiO2–CaO–P2O5–CeO2 System"

_gels, 2022, doi:10.3390/gels8060344_

Round 1

Reviewer 1 Report

This manuscript deals with the analysis of the crystallization behavior of tree mesoporous bioglasses in the system 70SiO2-(26-x)CaO-4P2O5-xCeO2. The data presented in this paper are interesting for the community working in this field and rather well described. They deserve being published however minor changes and corrections in the paper can be useful.

The introduction is rich with a large number of information and quotations, however some information are not directly relevant for the paper and could be remove. And some more general information about mesoporous bioglasses could be added, especially about the evaporation induced self assembly method. One wonder also what is the effect of mesoporosity on crystallization?

The structure of the gels obtained by sol_gel method coupled with evaporation induced self assembly method is first studied by IR and Raman spectroscopy. Unfortunately, the main conclusions of these analysis are not clearly established…

These gels are further analyzed by ATD/DTG/TG. The data and interpretation are interesting but more information about the mesostructure and the pores evolution would have been valuable. Moreover, I didn’t understand why the author attribute the last event in the DTG graph to a glass transition. A glass transition is not associated to a mass loss, but it should be associated with an event on DTA graph…?

There is a problem with the number 2.2 of the next paragraph (already used)…

This paragraph deals with the phase identification in the devitrified Ce containing MBG’s.

Lines 171 to 174, it is not clear: why crystalline phosphate are observed? Couldn’t it be amorphous PO43+ groups? Could the authors give more explanation about that?

In this page and the following, there is a lot of spaces missing between words, like line 171 “BGswith” or line 172 “5by”, and so on…

The next paragraph deals with the crystallization behavior of the stabilized MBGs. The estimation of the processing range of the glass is estimated as Tx-Tg. Tg is explained to defined as the inflexion point on the DSC curve. However nothing is said about the difficulty of Tg determination. Measurement uncertainties would have been valuable associated with these data, as well as a discussion about Tg  and Tx determinations.

The last paragraph concerns the bioactivity of the MBG5Ce_T samples.

Figure 10 should be described and discussed.

Line 350 “size below 75mm” shoudn’t it be “below 75µm”?

Although I am not an english native, I find that for some entences english can be improved.

Author Response

Dear Reviewer,

We are very grateful for your valuable comments enabling us to improve our manuscript. Herewith you find the responses for each raised issue.

Reviewer comments are shown in black font.

Authors’ responses are shown in blue font.

Revised texts are green highlighted in the manuscript.

Reviewer 2 Report

This is a very comprehensive paper regarding the crystallization study of some mesoporous bioactive glasses based on SiO2–CaO-3P2O5-CeO2 system.  In the introduction, the authors briefly present the reason why this feature of bioglases is being studied. Also, the techniques used are suitable for the study of the crystallization process.

I agree with the publication of the manuscript, but few issues should be improved:

- There are some abbreviations that are not defined the first time they appear in the text, such as “70S30C” (Line 78), “70S26C4P” (Line 144), “58S” (Line 185)… I recommend explaining the composition of these bioglasses in parentheses, for a better understanding by readers.

- Line 56: Specify what represent the ratios: “(1:98 and 13:93)”?

- I recommend paying attention when referring to figures in the text, for example:

Line 82 – “Figure 2a” - I think the authors wanted to refer to Figure 1a, because it's about the bands in the IR spectrum of G5Ce sample;

Line 93 – “Figure 2b” - is in fact Figure 1b (UV-Raman spectra of the GxCe gels), not DTA curves.

Line 122: “Figures 1b and 3a” – In the text, the authors talk about the DTG curve for the G1Ce, so it's about the Figure 2a.

- In Table 1, replace “Moist” with “Moisture” or better with “Physically absorbed water”, to be the same as in sample "0Ce";

- Activation energy is abbreviated: “Ec” (Line 234), “EC” (Line 356) and “Ea” (Line 360). Which one is correct? Use only one abbreviation;

- Line 279 – “MBG5Ce_(RT/830/878) samples”; line 282 – “MBG5Ce_(830/875)_14”; line 289 – “MBG5Ce_(830/876)_14 samples”. Which temperature is correct? 878, 875 or 876 for MBG5Ce sample? In Figure 9a,b it’s about 876. Instead, in Figure 9 caption it's about 875? So which one is correct?

- Figure 10b: What does the inset graph represent? It is not mentioned at all in the text. Why only in figure 10b and not in the others?

Author Response

(The authors gave the same response as above.)
